# ExRL: Sample-Efficient Online Adaptation of Action Chunking Policies via Execution Length Control

Xu Luo[1], Jiaying Yang[2], Hao Wu[1], Junlin Xie[1], Zijin Hong[3], Ji Zhang[1], Lianli Gao[1], Jingkuan Song[2]

[1]UESTC     [2]Tongji University     [3]The Hong Kong Polytechnic University

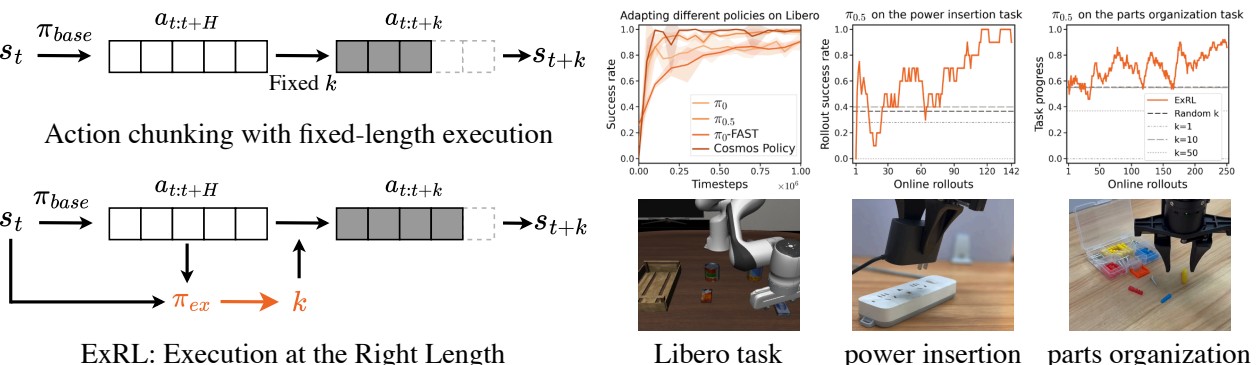

Fig. 1: **ExRL uses execution length control for sample-efficient online adaptation of action chunking policies.** *Left*: While standard action chunking executes a fixed-length prefix of each predicted action chunk, ExRL uses reinforcement learning to train a lightweight execution-length policy that dynamically selects the prefix length for each chunk. *Right*: ExRL achieves sample-efficient online adaptation on both simulation and real-world tasks; it improves different classes of generalist robot policies on a `Libero` task, and adapts $\pi_{0.5}$ to two challenging real-world manipulation tasks, including power insertion that demands high precision, and parts organization which is a long-horizon, dexterous task.

*Abstract*—**Robot policies with action chunking learned from human demonstrations provide a strong starting point for general and precise robotic control, but often still require further adaptation to perform robustly in the real world. Reinforcement learning offers a promising route to autonomous policy improvement, but existing approaches can require substantial online interaction and are often tied to particular policy classes or parameterizations. We present *Execution at the Right Length* (ExRL), a lightweight method for sample-efficient online adaptation of various action chunking policies. Unlike previous methods that adapt *what actions* the policy predicts, ExRL adapts *how long* to execute each predicted action chunk before querying the policy again. With the base policy frozen, a lightweight critic (and optionally an actor) is used to select this execution length online, reducing adaptation from continuous, high-dimensional action optimization to a compact discrete decision about when to replan, which enables sample-efficient adaptation while still yielding substantial gains. Because it is added on top of a frozen policy, ExRL is inherently compatible with diverse policy classes and backbones. We validate ExRL on simulation benchmarks and real-world robotic tasks, showing sample-efficient adaptation across a range of policies, including pretrained vision-language-action models and a world action model, demonstrating its broad utility for real-world policy improvement.**

## I. INTRODUCTION

Robot learning has made substantial progress in robotic control by reducing reliance on manually engineered, task-specific controllers and instead learning policies directly from data. In particular, action chunking policies trained on human demonstration datasets, which predict action sequences at each query [67], have enabled robots to perform increasingly complex behaviors across a wide range of tasks [53, 5, 6]. However, these policies are rarely optimal when deployed in the real world, where limited demonstration coverage and imperfect generalization can cause small execution errors to compound and bring the robot into states where the policy becomes unreliable [49]. In such cases, utilizing online experience collected during deployment to improve the policy becomes important.

Reinforcement learning (RL) provides a direct way to learn from online experience. In other domains such as language modeling, online RL has been shown to be quite effective for adapting models pretrained on offline datasets [40, 23, 18]. For robotic control, however, online data collection is often slow, costly, and sometimes unsafe, making sample efficiency a central requirement. Meeting this requirement is particularly

difficult when RL is used to adapt action chunking policies, since optimizing over a continuous, high-dimensional space of action chunks can require many online trials. In addition, recent robot policies differ widely in their action parameterizations and architectures [67, 11, 26, 6, 28], but existing RL algorithms are often designed for specific policy classes [38, 56, 62]. This makes them difficult to transfer across backbones without redesigning key components.

In this work, we take a different perspective to tackle these challenges. Rather than directly adapting the high-dimensional action chunk predicted by the policy, we make the key observation that action chunking policies expose another adaptation interface that is much smaller and broadly shared across policy classes, namely, how long to execute a predicted action chunk before querying the policy again. While usually fixed during deployment, this execution length has been shown to strongly influence policy behavior [48, 65]. Executing multiple actions open-loop can preserve the temporal structure of a predicted chunk [35] and play an important role in mitigating compounding errors [65]. At the same time, replanning after fewer actions gives the policy more frequent access to recent observations, making it more reactive when the robot encounters unexpected state changes or dynamics mismatch [35]. Given the complexity of environment dynamics and the variability in the policy's predicted action chunks, the optimal execution length may vary across tasks, states, and policy predictions. We thus conjecture that policy improvement can be achieved by dynamically controlling this execution length based on the current state and the predicted action chunk.

We instantiate this idea as *Execution at the Right Length* (ExRL), a sample-efficient online RL method for adapting action chunking policies (Figure 1). It keeps the base policy frozen and adds only a lightweight execution-length policy that selects how many actions to execute from each predicted chunk before replanning. In this way, online adaptation is reduced from optimizing continuous, high-dimensional action chunks to learning over a compact discrete decision space, enabling efficient learning from limited online interaction. The discrete nature of the execution-length decision also makes it possible to use actor-free value-based RL algorithms [59, 39] for robotic policy adaptation, which we show may further increase sample efficiency. Because ExRL does not modify the parameters of the base policy, it can be applied on top of virtually any classes of action chunking policies. This policy-agnostic interface even makes it possible to transfer across backbones, allowing an adapter learned with one base policy to potentially improve another.

Our main contribution is ExRL, a simple but effective method for efficient online adaptation of various classes of action chunking policies. The use of execution length as the adaptation interface makes our approach lightweight and broadly applicable. We empirically demonstrate the effectiveness of ExRL across simulated and real-world manipulation tasks, showing state-of-the-art online adaptation performance across a range of action chunking policies, including different classes of pretrained vision-language-action (VLA) models

such as $\pi_{0.5}$ [22] and $\pi_0$-FAST [43], as well as a world action model, Cosmos Policy [28]. On challenging real-world tasks, including precise manipulation and long-horizon tasks, ExRL enables sample-efficient adaptation of the $\pi_{0.5}$ policy within a few hundred online trials, improving average task progress from below $50\%$ to above $80\%$.

## II. RELATED WORK

**Imitation learning in robotics.** Imitation learning, particularly behavior cloning, has been widely used to train data-driven policies for robotic control [3, 4, 45, 66, 16, 46]. Prior work has explored a wide range of action parameterizations and policy architectures for imitation learning. Earlier methods often represent actions with Gaussian or GMM policies [24, 37], while more recent approaches use expressive generative modeling techniques, including VAEs [67], VQ-VAEs [29], diffusion or flow models [11, 54, 6, 5], and autoregressive next-token prediction models [68, 26, 43]. With the growing scale of human demonstration datasets and policy models [57, 25, 8, 41], imitation learning has further become a pretraining paradigm for generalist robot policies, absorbing broad prior knowledge into a single model that can be adapted for real-world deployment [54, 26, 6]. Our work provides an efficient method to adapt these diverse policy classes from online experience.

**Adaptive execution of action chunks.** Proposed by Zhao et al. [67], action chunking has become a de facto design choice in robot policies learned with imitation learning. Recognizing that the execution length of action chunks strongly affects policy behavior [48, 65, 35], several concurrent works have explored adaptive execution strategies. AAC [33] selects the execution length of vision-language-action models using action entropy, while FFDC [58] selects the execution length for world action models by verifying whether predicted future observations remain consistent with the real rollout. Compared to our work, these methods rely on heuristic surrogates and cannot benefit from online experience, which can limit their performance improvement. Two other concurrent works, AC-SAC [10] and ACH [47], learn value-based adaptive chunk-size selection for offline-to-online RL. However, these methods train chunked policies within a specific RL framework, and are not designed as plug-in adapters for arbitrary pretrained action chunking policies.

**Reinforcement learning for robot policy adaptation.** Similar to its role in language modeling, RL is considered to be an effective tool for adapting robot policies from online experience. On-policy methods such as proximal policy optimization (PPO) have been used to fine-tune pretrained robot policies [31, 36, 9, 44], but they often require many online rollouts, making them difficult to extend to real-world RL. Off-policy RL algorithms such as SAC [19] and TD3 [17] can use data more efficiently, but they assume deterministic or Gaussian policy parameterizations, and are thus difficult to directly apply to generative policies with complex action distributions. Several methods address this challenge by adapting specific

policy classes: DSRL [56] and ZPRL [63] optimize diffusion/flow policies in latent spaces, while OGPO [42] and RL-100 [30] adapt diffusion policies by reformulating the denoising process as an unrolled bi-level MDP. These methods can improve sample efficiency, but remain tied to particular policy classes. Another line of work studies residual RL, which learns a residual policy that additively modifies the actions produced by the base policy [64, 2, 1, 13, 60]. While policy-agnostic and lightweight, Residual RL still optimizes continuous, high-dimensional action corrections, making it less sample-efficient for adapting action chunking policies. PA-RL [38] achieves policy-agnostic adaptation by casting RL as action optimization followed by supervised policy learning. RECAP [21] builds an offline-to-online pipeline for training advantage-conditioned policies, which can incorporate batch online RL [14] for efficient adaptation. In contrast to these action-modifying approaches, ExRL keeps the base policy frozen and adapts only the execution length of each predicted chunk, reducing online adaptation to a compact discrete decision space and enabling sample-efficient, policy-agnostic adaptation.

## III. PRELIMINARIES

**Markov decision processes.** Markov decision process (MDP) is denoted by a tuple $\mathcal{M} = (\mathcal{S}, \mathcal{A}, P, r, \rho_0, \gamma)$, where $\mathcal{S}$ is the state space, $\mathcal{A}$ is the action space, $P(\cdot \mid s, a)$ is the transition dynamics, $r : \mathcal{S} \times \mathcal{A} \to \mathbb{R}$ is the reward function, $\rho_0$ is the initial state distribution, and $\gamma \in [0, 1)$ is the discount factor. A policy $\pi : \mathcal{S} \to \Delta(\mathcal{A})$ is a mapping from states to distributions over actions. The $Q$-function, or critic, of a policy $\pi$ is defined as $Q^\pi(s, a) = \mathbb{E}_\pi[\sum_{t \geq 0} \gamma^t r(s_t, a_t) \mid s_0 = s, a_0 = a]$, which is the expected discounted return of $\pi$ from taking action $a$ in state $s$. The goal of reinforcement learning is to find a policy that maximizes the expected discounted return.

**Q-learning.** A broad class of RL algorithms learns a parameterized Q-function $Q_\phi(s_t, a_t)$ by bootstrapping from Bellman targets. Given a transition $(s_t, a_t, r_t, s_{t+1})$, a generic one-step temporal-difference (TD) target takes the form

$$y_t = r_t + \gamma V(s_{t+1}), \tag{1}$$

where $V$ denotes the bootstrap value of the next decision state induced by the particular algorithm. The critic is then trained by minimizing the squared TD error

$$\min_\phi \ \mathbb{E}\left[(Q_\phi(s_t, a_t) - y_t)^2\right]. \tag{2}$$

**Action chunking.** An action chunking policy $\pi$ is a mapping $\mathcal{S} \to \Delta(\mathcal{A}^H)$ from states to distributions over $H$-step action sequences. At a query state $s_t$, a standard action chunking policy samples an action chunk $a_{t:t+H} = [a_t, a_{t+1}, ..., a_{t+H-1}] \sim \pi(\cdot \mid s_t)$, and then executes $a_{t:t+k}$, where $1 \leq k \leq H$ is a fixed execution length, after which the policy is queried again at state $s_{t+k}$.

**Problem setting.** In this paper, we assume access to a pretrained action chunking policy $\pi_{\text{base}}$, and our goal is to adapt it to maximize the expected return specified by some

reward $r$ using online interactions with an environment $\mathcal{M}$. Note that we make no assumptions on the parameterization or architecture of $\pi_{\text{base}}$.

## IV. EXECUTION AT THE RIGHT LENGTH

We propose ExRL, which adapts action chunking policies by dynamically controlling how long each predicted action chunk is executed before replanning, rather than modifying the action chunk itself (Figure 1). In this section, we formalize this idea as reinforcement learning over execution-length decisions.

Given a pretrained action chunking policy $\pi_{\text{base}}$ with chunk size $H$, we freeze $\pi_{\text{base}}$ and learn an execution-length policy $\pi_{\text{ex}} : \mathcal{S} \times \mathcal{A}^H \to \Delta(\mathcal{K})$, where $\mathcal{K} = \{1, \ldots, H\}$. At each query, $\pi_{\text{base}}$ first predicts an action chunk, and $\pi_{\text{ex}}$ maps the current state and this predicted chunk to a distribution over feasible execution lengths. In this way, the raw primitive actions are still produced by $\pi_{\text{base}}$, while the learned execution-length policy acts in the much smaller discrete space $\mathcal{K}$, avoiding direct optimization in the continuous, high-dimensional space of action chunks. We can also optionally allow the policy to choose $k = 0$, corresponding to executing none of the actions in the current chunk, which is essentially an immediate replanning decision.

**Duration-aware Q-learning.** The execution-length decision naturally leads to a temporally extended Q-function. At a query timestep $t$, let $a_{t:t+H}$ denote the action chunk sampled from $\pi_{\text{base}}(\cdot \mid s_t)$. In ExRL, we learn a critic $Q_\phi(s_t, a_{t:t+H}, k)$, which estimates the return obtained by executing the first $k$ actions of the predicted chunk and then continuing with the current execution-length strategy. This is closely related to Q-chunking [32], which learns a critic over an action sequence rather than a single primitive action. The difference is that in Q-chunking, the critic evaluates the action chunks, while the critic in ExRL evaluates the selected prefix lengths of action chunks. Thus, the accumulated reward and the bootstrap discount in ExRL both depend on the chosen "meta-action" $k_t$, leading to the duration-aware TD target below:

$$y_t = \sum_{i=0}^{k_t-1} \gamma^i r(s_{t+i}, a_{t+i}) + \gamma^{k_t} V(s', a'), \tag{3}$$

where $s' = s_{t+k_t}$ is the next query state, $a' = a_{t+k_t:t+k_t+H} \sim \pi_{\text{base}}(\cdot \mid s')$ is the next chunk sampled from the frozen base policy, and $V$ is the next-state value. This duration-aware backup mirrors the Bellman backup used in semi-Markov decision processes (SMDPs) [52], where temporally extended decisions may last for a variable number of primitive timesteps. In Appendix A, we formalize this connection by showing how execution-length control induces an SMDP under the frozen base policy $\pi_{\text{base}}$.

**Practical instantiations.** ExRL is agnostic to the choice of RL algorithm, as long as the algorithm can optimize over the discrete execution-length space with the duration-aware backup above. In practice, we instantiate ExRL with two off-policy variants for sample-efficient online adaptation. The first variant, **ExRL-Q**, utilizes the discrete nature of the action

**Algorithm 1** Execution-Length Control via Q-Learning (ExRL-Q)

1: **input:** pretrained action chunking policy $\pi_{\text{base}}$, online environment $\mathcal{M}$, execution-length space $\mathcal{K} = \{1, \ldots, H\}$
2: Initialize replay buffer $\mathcal{D}_{\text{replay}}$
3: Initialize execution-length critic $Q_\phi$
4: Initialize target critic $Q_{\bar\phi}$ with $\bar\phi \leftarrow \phi$
5: **for** each training iteration **do**
6:     ▷ Online Rollout
7:     Sample an action chunk $a_{t:t+H} \sim \pi_{\text{base}}(\cdot \mid s_t)$ at the current query state $s_t$
8:     Select execution length $k_t$ with Equation (6)
9:     Execute the first $k_t$ actions in the chunk and observe $\hat{r}_t \leftarrow \sum_{i=0}^{k_t-1} \gamma^i r(s_{t+i}, a_{t+i})$ and $s' \leftarrow s_{t+k_t}$
10:     Add $(s_t, a_{t:t+H}, k_t, \hat{r}_t, s')$ to $\mathcal{D}_{\text{replay}}$
11:     ▷ Train Critic
12:     Sample $(s, a, k, \hat{r}, s') \sim \mathcal{D}_{\text{replay}}$ and $a' \sim \pi_{\text{base}}(\cdot \mid s')$
13:     Compute next-state value $V(s', a')$ with Equation (4)
14:     Compute TD target $y_t$ with Equation (1)
15:     Compute TD loss $\mathcal{L}_Q(\phi)$ with Equation (5)
16:     Train critic $Q_\phi$ by minimizing $\mathcal{L}_Q(\phi)$
17:     Periodically update target: $\bar\phi \leftarrow \tau\phi + (1-\tau)\bar\phi$
18: **return** $Q_\phi$

**Algorithm 2** Execution-Length Control via Discrete Soft Actor-Critic (ExRL-SAC)

1: **input:** pretrained action chunking policy $\pi_{\text{base}}$, online environment $\mathcal{M}$, execution-length space $\mathcal{K} = \{1, \ldots, H\}$, temperature $\alpha$
2: Initialize replay buffer $\mathcal{D}_{\text{replay}}$
3: Initialize execution-length critic $Q_\phi$
4: Initialize target critic $Q_{\bar\phi}$ with $\bar\phi \leftarrow \phi$
5: Initialize execution-length actor $\pi_\psi$
6: **for** each training iteration **do**
7:     ▷ Online Rollout
8:     Sample an action chunk $a_{t:t+H} \sim \pi_{\text{base}}(\cdot \mid s_t)$ at the current query state $s_t$
9:     Sample execution length $k_t \sim \pi_\psi(\cdot \mid s_t, a_{t:t+H})$
10:     Execute the first $k_t$ actions in the chunk and observe $\hat{r}_t \leftarrow \sum_{i=0}^{k_t-1} \gamma^i r(s_{t+i}, a_{t+i})$ and $s' \leftarrow s_{t+k_t}$
11:     Add $(s_t, a_{t:t+H}, k_t, \hat{r}_t, s')$ to $\mathcal{D}_{\text{replay}}$
12:     ▷ Train Critic and Actor
13:     Sample $(s, a, k, \hat{r}, s') \sim \mathcal{D}_{\text{replay}}$ and $a' \sim \pi_{\text{base}}(\cdot \mid s')$
14:     Compute the soft value $V(s', a')$ with Equation (7)
15:     Compute TD target $y_t$ with Equation (1)
16:     Compute TD loss $\mathcal{L}_Q(\phi)$ with Equation (5)
17:     Compute actor loss $\mathcal{L}_\pi(\psi)$ with Equation (8)
18:     Train critic $Q_\phi$ by minimizing $\mathcal{L}_Q(\phi)$
19:     Train actor $\pi_\psi$ by minimizing $\mathcal{L}_\pi(\psi)$
20:     Periodically update target: $\bar\phi \leftarrow \tau\phi + (1-\tau)\bar\phi$
21: **return** $\pi_\psi$

space and uses value-based RL without learning an explicit actor [59, 39]. In this case, the next-state value $V$ is computed by a Bellman optimality backup:

$$V(s', a') = \max_{k' \in \mathcal{K}} Q_{\bar\phi}(s', a', k'), \tag{4}$$

where $Q_{\bar\phi}$ is a slowly-updated target, and the critic is trained by minimizing the squared TD error:

$$\mathcal{L}_Q(\phi) = \mathbb{E}\left[ (Q_\phi(s_t, a_{t:t+H}, k_t) - y_t)^2 \right]. \tag{5}$$

During online rollout, the execution length is selected greedily as

$$k_t = \arg\max_{k \in \mathcal{K}} Q_\phi(s_t, a_{t:t+H}, k). \tag{6}$$

The second variant, **ExRL-SAC**, adds an explicit actor $\pi_\psi(k_t \mid s_t, a_{t:t+H})$ that outputs a categorical distribution over execution lengths. Since $\mathcal{K}$ is discrete, we use discrete SAC [12] as an actor-critic instantiation. Compared with continuous-action SAC, the discrete action space allows us to compute the policy value exactly by summing over all $k \in \mathcal{K}$. In this case, the hard maximum in ExRL-Q is replaced by the soft value under the actor. The soft value is

$$V(s', a') = \sum_{k' \in \mathcal{K}} \pi_\psi(k' \mid s', a')\Big[ Q_{\bar\phi}(s', a', k') \\ -\alpha \log \pi_\psi(k' \mid s', a') \Big]. \tag{7}$$

The entropy term $-\alpha \log \pi_\psi$ encourages the policy to maintain uncertainty over execution lengths, which improves exploration and prevents early collapse to a single fixed $k$.

Squared TD error is used to update the critic, and the actor is trained to choose execution length with high soft value:

$$\mathcal{L}_\pi(\psi) = -\mathbb{E}_{k_t \sim \pi_\psi(\cdot \mid s_t, a_{t:t+H})}\Big[ Q_\phi(s_t, a_{t:t+H}, k_t) \\ -\alpha \log \pi_\psi(k_t \mid s_t, a_{t:t+H}) \Big]. \tag{8}$$

The algorithms of ExRL-Q and ExRL-SAC are summarized in Algorithm 1 and 2, respectively.

## V. EXPERIMENTS

The goal of our experiments is to answer the following core questions:

(**Q1**) Can ExRL enable sample-efficient adaptation across standard action chunking policies and generalist robot policies in both simulation and real-world tasks?
(**Q2**) Can the learned execution-length policy transfer from one base policy to another?
(**Q3**) What execution-length patterns does ExRL learn?
(**Q4**) How do base policy chunk size and key design choices of ExRL affect adaptation performance?

### A. Base Policies

To show the generality of our method, we conduct experiments on diverse classes of action chunking policies. We consider both representative action chunking policies trained

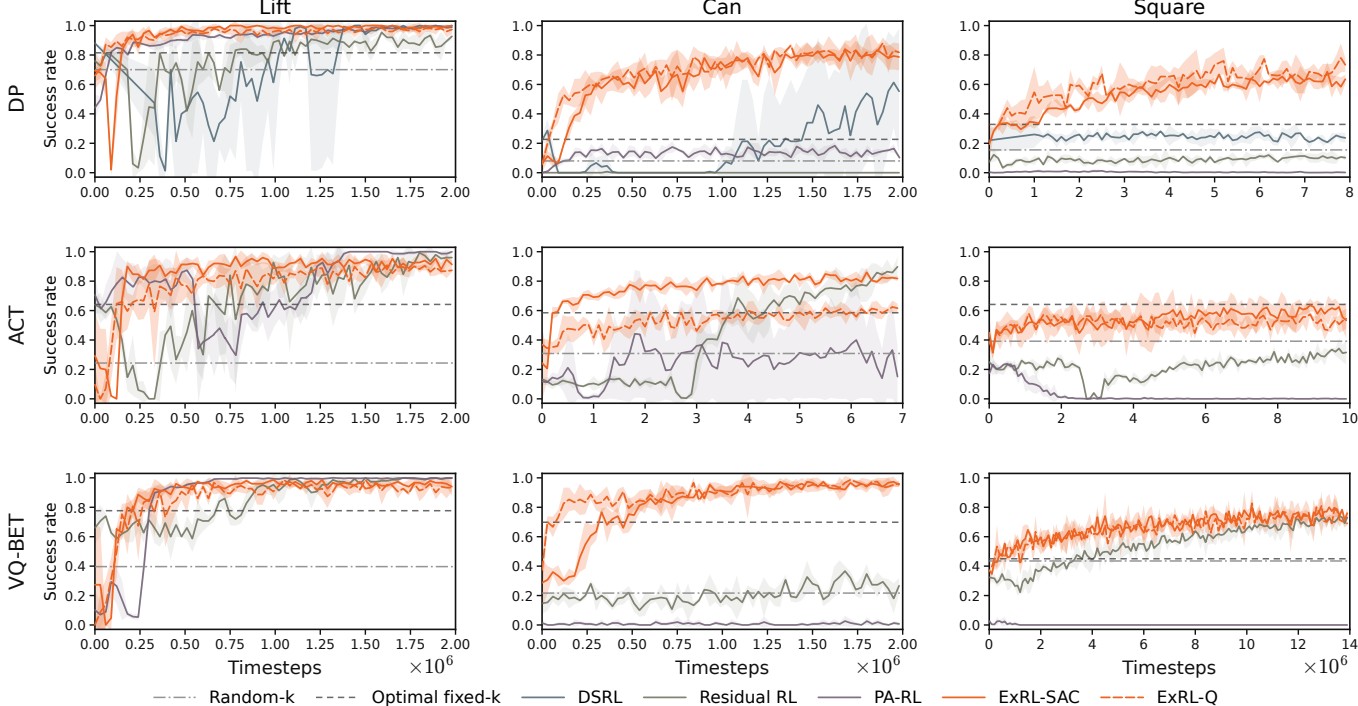

Fig. 2: ExRL enables sample-efficient online adaptation of various action chunking policies, including DP, ACT, and VQ-BeT, on `Robomimic`.

with imitation learning and several pretrained generalist robot policies of different classes.

**Representative action chunking policies**: (1) **Diffusion Policy (DP)** [11]: an action chunking policy that models multimodal continuous action chunks using denoising diffusion [20]; (2) **ACT** [67]: a transformer-based chunking policy with a conditional VAE [50] latent variable for modeling multimodal action chunks; and (3) **VQ-BeT** [29]: a behavior transformer that uses VQ-VAE [55] latent action codes to represent multimodal action chunks.

**Pretrained generalist robot policies.** (1) $\pi_0$ [6] and $\pi_{0.5}$ [22]: state-of-the-art VLA models that use flow matching [34] to model action chunks; (2) $\pi_0$-**FAST** [43]: an autoregressive VLA model that generates action chunks through efficient action tokenization; and (3) **Cosmos Policy** [28]: a world action model built on a large pretrained video generation model.

### B. Baselines

We compare ExRL against execution-length baselines and three strong online adaptation baselines, with a focus on policy-agnostic methods for action chunking policies. We omit methods that are either not directly applicable to the policy classes considered here or have been shown in prior work to underperform these stronger baselines.

**Execution-length baselines.** (1) **Random-$k$**: randomly samples an execution length from the feasible set at each policy query; (2) **Optimal fixed-$k$**: The optimal fixed-length execution baseline. We sweep possible length candidates and report the best-performing one.

**DSRL** [56]: A state-of-the-art method for sample-efficient adaptation of diffusion- or flow-based policies. It keeps the base policy frozen and learns to steer its sampling latent, such as the initial noise of a diffusion or flow policy, toward higher-return actions;

**Residual RL** [64]: A simple, flexible, and policy-agnostic adaptation approach that serves as a common building block in many policy refinement methods [64, 2, 1, 13, 60, 15, 51]. It learns an additive residual correction on top of the action chunks predicted by the frozen base policy. We use the standard implementation from Policy Decorator [64].

**PA-RL** [38]: A policy-agnostic RL finetuning method that first optimizes candidates of action chunks predicted by the base policy using critic-guided re-ranking and gradient-based local improvement, and then distills the optimized actions back into the policy with a supervised learning loss. While PA-RL is primarily used for offline-to-online fine-tuning, we use its pure online variant for a fair comparison.

### C. ExRL Enables Efficient Adaptation Across Various Classes of Action Chunking Policies

We first evaluate whether ExRL can provide sample-efficient online adaptation across representative action chunking policies. We evaluate on `Robomimic`, a challenging robotic manipulation benchmark. For each of the three tasks, `Lift`, `Can`, and `Square`, we train DP, ACT, and VQ-BeT models with chunk sizes of 50 and 32 (See Appendix V-G for ablations on chunk size). We report the average success rate

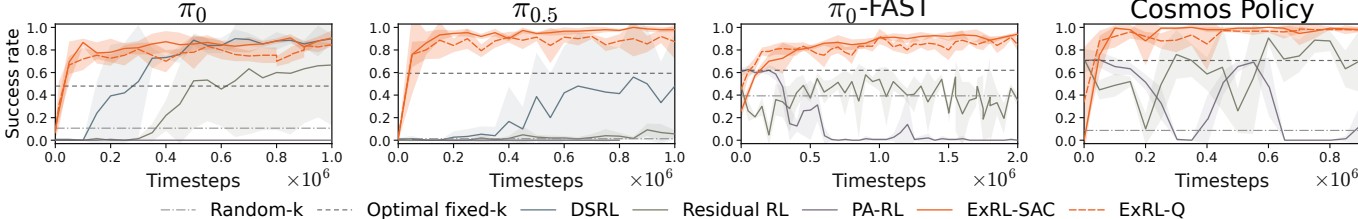

Fig. 3: ExRL enables sample-efficient online adaptation of different classes of generalist robot policies on `Libero`, including the flow-based VLA models $\pi_0$ and $\pi_{0.5}$, the autoregressive VLA model $\pi_0$-FAST, and the world action model Cosmos Policy.

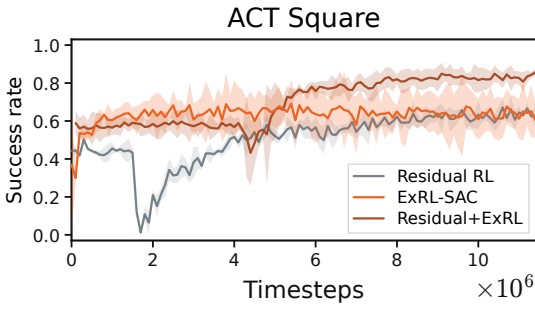

Fig. 4: ExRL can be combined with Residual RL to improve final performance while being more sample-efficient than Residual RL alone.

| Method | Power insertion | Parts organization |
|---|---|---|
| Random $k$ | 11/30 | 83/150 |
| Fixed $k = 1$ | 7/25 | 0/125 |
| Fixed $k = 10$ | 10/25 | 69/125 |
| Fixed $k = 50$ | 0/25 | 46/125 |
| ExRL-Q | **22/25** | **104/125** |

TABLE I: Real-world adaptation of $\pi_{0.5}$ with ExRL. Power insertion reports successful trials, while parts organization reports task progress as total completed stages.

and 95% confidence intervals over three seeds.

We present the results in Figure 2, where chunk size is 50 for DP and ACT, and 32 for VQ-BeT. We also prsent results of chunk size 32 for DP and ACT in Figure 11. First, we note that while the optimal fixed-$k$ baseline consistently outperforms the random-$k$ baseline, ExRL at most of the time achieves much higher performance across all three policy classes. As ExRL improves the policy solely by changing how long each predicted chunk is executed without modifying the predicted actions of the base policy, this validates the importance and expressiveness of execution-length control as an adaptation interface. Compared to other RL baselines, ExRL achieves strong sample efficiency for adapting action chunking policies, often at a speed much faster at the beginning. This advantage is especially clear on the more challenging `Can` and `Square` tasks, where Residual RL, DSRL, and PA-RL often improve more slowly, validating the advantage of adoption of a small, discrete optimization space over continuous, high-dimensional (latent) action spaces. Both ExRL-SAC and ExRL-Q perform well, showing that execution length control can be learned effectively with either an actor-critic method or an actor-free value-based method.

While ExRL is sample-efficient, its performance can sometimes be bounded by the action distribution of the frozen base policy, as in the `Square` task. Figure 4 shows that this limitation can be alleviated by combining ExRL with Residual RL, which allows the adapted policy to move beyond the action chunks proposed by the frozen base policy. In this experiment, we first train ExRL for a number of steps, then freeze the ExRL policy and train a Residual RL policy on top. This simple recipe improves the performance upper bound while remaining more sample-efficient than Residual RL alone, with ExRL serving as a prior for residual RL.

*D. Adapting Different Classes of Generalist Robot Policies with ExRL*

We next evaluate whether ExRL can adapt pretrained generalist policies, which are much larger in size and produce higher-dimensional action chunks. Following the setting of DSRL, we adapt the four generalist policies from different classes described above to a pick-and-place task from the `LIBERO-90` suite. We first fine-tune each pretrained policy on this task, and then adapt it with online RL. The results are shown in Figure 3. ExRL substantially improves all four pretrained generalist policies. For example, it improves $\pi_{0.5}$ from nearly 0% to 90% within only 200,000 online steps, and achieves similar gains on $\pi_0$, $\pi_0$-FAST, and Cosmos Policy. In contrast, action- or latent-space adaptation baselines either improve more slowly or fail to consistently improve across policy classes. This setting is challenging for online RL due to a combination of large model sizes, high-dimensional action chunks, long horizons, visual observations, and sparse rewards. These results highlight the advantage of adapting the compact execution-length space. To the best of our knowledge, ExRL is the first RL method to successfully adapt flow-based VLAs, autoregressive VLAs, and world action models within a unified framework.

We further evaluate ExRL on adapting $\pi_{0.5}$ to two challenging real-world tasks. The first task, *power insertion*, requires inserting a plug into a power socket and is challenging due to precise contact-rich manipulation. The second task, *parts organization*, requires placing small parts into their

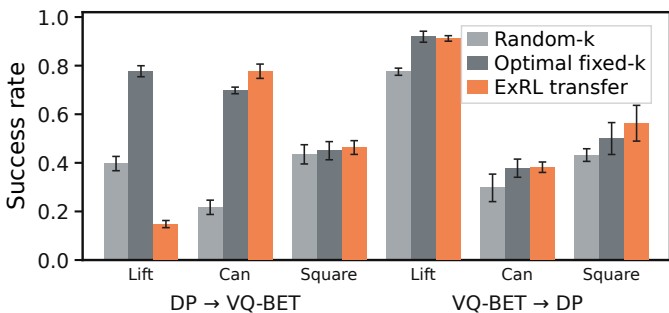

Fig. 5: Transfer of learned execution-length policies across base policies.

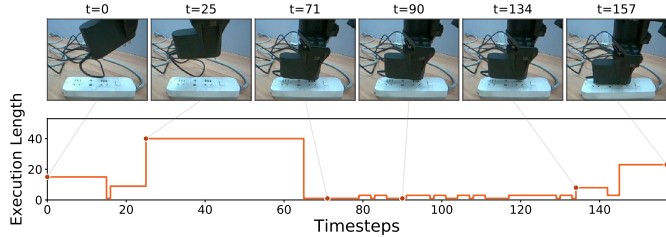

Fig. 6: **A representative execution-length pattern learned by ExRL.** We demonstrate a typical power insertion rollout. ExRL uses longer executions for coarse motion, shorter executions for precise alignment, and longer executions again when the base policy predicts a reliable insertion chunk.

corresponding locations in a storage box and then closing the box. This task is challenging because it is long-horizon, involves small objects, and requires reliable recovery across stages. We treat each successful placement and the final box closing as one completed stage. We report success rate for power insertion and task progress, measured by the number of completed stages, for parts organization. As shown in Figure 1 and Table I, ExRL-Q improves $\pi_{0.5}$ on both challenging tasks within a few hundred real-world rollouts, leading to adaptation within a few hours. After training, ExRL-Q achieves a high success rate on power insertion and complete most stages on parts organization, demonstrating sample-efficient real-world adaptation for both precise contact-rich and long-horizon manipulation tasks.

### E. Can Execution-Length Policies Transfer Across Base Policies?

We next study whether an execution-length policy $\pi_{ex}$ learned on one base policy $\pi_{base}^a$ can be reused for another base policy $\pi_{base}^b$ on the same task. Since ExRL adapts only the replanning decision rather than the base policy itself, such transfer is possible when different base policies induce similar useful chunk-execution patterns.

We demonstrate this with DP and VQ-BeT in Figure 5: an ExRL policy trained on one of them can be directly applied to the other and still provide non-trivial improvement over random execution lengths and, in some cases, outperform the optimal fixed-$k$ baseline. This transfer is not universal. We find that transferred execution-length policies can fail, e.g., DP transferred to VQ-BeT on Lift. We hypothesize that transfer depends on similarities in the demonstration data, the error distribution within action chunks, and how each policy fits temporal correlations within chunks. To the best of our knowledge, this is the first demonstration of an RL-trained adaptation module that can be transferred across different base policies.

### F. What Execution Length Patterns Does ExRL Learn?

To understand the behavior learned by ExRL, we visualize a typical real-world rollout on *power insertion* in Figure 6. ExRL tends to use longer execution lengths during coarse motion, then switches to short executions during precise alignment

near contact. When the base policy predicts a reliable insertion chunk, ExRL commits to a longer execution again to complete the insertion. This behavior suggests that ExRL learns when to trust the predicted chunk rather than simply choosing short or long execution lengths. Short executions provide high reactivity, but near contact the policy may receive visually similar neighboring states and repeatedly output similar small motions, which can prevent progress or push the robot out of distribution. ExRL often waits with short executions until a useful insertion or correction chunk appears, and then executes it for longer. This illustrates how execution-length control can coordinate replanning and open-loop execution in a state- and chunk-dependent way.

### G. On the Influence of Base Policy Chunk Size on Adaptation Performance

We investigate how the chunk size of the base policy affects the adaptation performance of ExRL and baseline methods such as DSRL. For all chunk sizes ranging from 2 to 32, we use the same base policy of chunk size 32 and mask some actions to turn it into a small-chunk policy. As shown in Figure 7, DSRL can be more sample-efficient when the base policy uses small chunks. However, as the chunk size increases, ExRL becomes increasingly advantageous. This is because a larger chunk provides a richer set of execution-length choices for ExRL, making execution-length control more expressive. At the same time, the optimization problem remains simple: even for long chunks, ExRL only optimizes over a small discrete set of at most tens of execution lengths. In contrast, action-space or latent-space adaptation methods such as DSRL optimize over a space whose dimension grows quickly with the action chunk. As the base policy chunk size increases, the latent or action space to be optimized becomes substantially larger, making online adaptation more difficult. Recent robot policies, especially generalist robot policies [22, 48, 27, 43, 28, 61], often use long action chunks to produce smoother behavior, reduce policy inference frequency, and preserve temporal structure within predicted actions. These results suggest that ExRL is especially well-suited to adapt these policies.

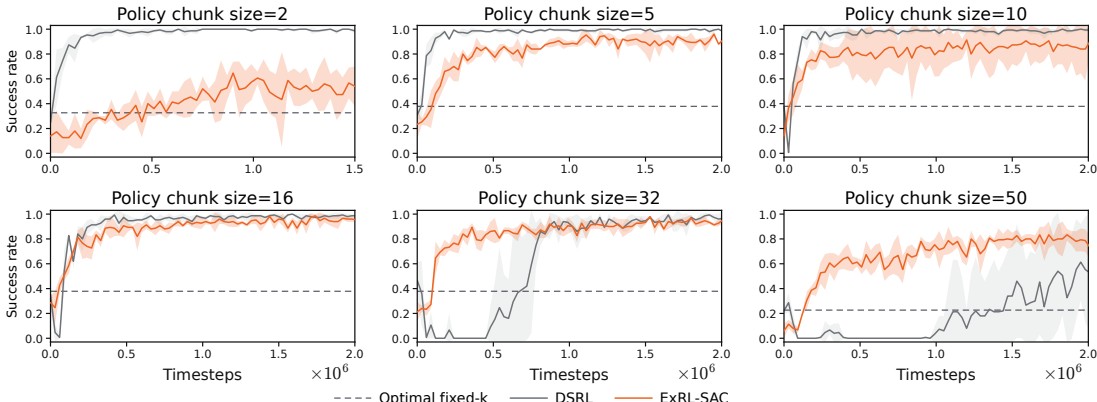

Fig. 7: **Comparison between ExRL and DSRL under different chunk sizes of base policy.** DSRL is more sample-efficient for policies with small chunk sizes. As the chunk size of base policy increases, ExRL has a larger space for execution-length optimization and becomes more advantageous.

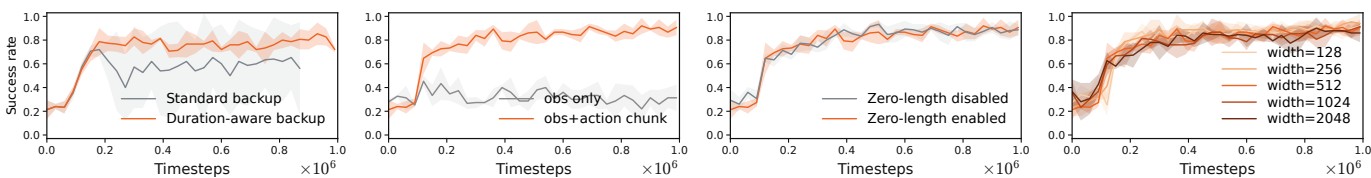

Fig. 8: **Ablation studies.** *Left to right*: Ablation on the duration-aware backup, inputs to the execution-length policy, zero-length execution choice, and network width. The duration-aware backup and conditioning on the predicted action chunk are important for effective execution-length learning, while allowing zero-length execution provides limited additional benefit and ExRL remains robust across network widths.

### H. Ablation Studies

We ablate several design choices in ExRL, including the duration-aware backup, the input to the execution-length policy, the zero-length action, and the network width of the execution-length policy. Results are shown in Figure 8.

**Duration-aware backup.** We first ablate the duration-aware TD target in ExRL. The duration-aware backup substantially improves performance, showing that correctly accounting for the temporal duration of each execution-length decision is important. Using a standard one-step backup ignores that different execution lengths correspond to different numbers of primitive timesteps, which can bias the critic toward shortcut choices such as consistently selecting overly short or overly long executions depending on the reward structure.

**Conditioning on the predicted action chunk.** We next compare an execution-length policy conditioned only on observations with one conditioned on both observations and the predicted action chunk. The result shows that conditioning on the action chunk is important, since the appropriate execution length depends not only on the current state but also on what the base policy intends to execute. Without access to the predicted chunk, the execution-length policy cannot reliably decide whether the current chunk should be trusted for longer open-loop execution or interrupted early for replanning.

**Zero-length execution.** We also ablate whether the execution-length policy can choose $k = 0$, which corresponds to rejecting

the current chunk and immediately replanning. On the tested task, enabling this option provides limited additional benefit, suggesting that choosing among positive execution lengths is often sufficient once the base policy produces reasonably useful chunks.

**Network width.** Finally, we vary the width of the execution-length policy network. ExRL remains performant across a wide range of widths, indicating that the method is not sensitive to this hyperparameter.

## VI. LIMITATIONS

ExRL has several limitations. First, like other sample-efficient methods that avoid learning directly in the raw action space, such as DSRL [56] and ZPRL [63], our method relies on the base policy to produce action chunks that are at least partially useful, and its performance is therefore bounded by the action distribution of the frozen base policy. Combining ExRL with action-space adaptation methods such as residual RL can alleviate this limitation, as shown in our experiments, but also increases algorithmic complexity. Second, like residual RL, ExRL does not modify the policy's action-generation process, and therefore cannot directly leverage offline human demonstration data. Third, our current experiments focus on single-task adaptation, and future work may consider language-conditioned or multi-task settings. Finally, ExRL may sometimes prefer shorter execution lengths to improve reactivity, which can slow down task completion due to increasing policy

inference time; incorporating time penalties into the reward or combining ExRL with asynchronous inference methods such as RTC [7] may help better balance reactivity and execution speed.

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

## APPENDIX

### A. Execution-Length Control as an Induced SMDP

In this section, we formalize the connection between ExRL and semi-Markov decision processes (SMDPs) [52]. Recall that the underlying environment is a primitive-time MDP $\mathcal{M} = (\mathcal{S}, \mathcal{A}, P, r, \rho_0, \gamma)$, where each primitive action advances the environment by one timestep. ExRL does not change this environment. Instead, by freezing an action chunking policy $\pi_{\text{base}}$ and learning only how many actions from each predicted chunk to execute, ExRL induces a higher-level decision process over execution-length decisions.

Let $\mathbf{a} = a_{t:t+H}$ denote an action chunk sampled from the frozen base policy at state $s$, i.e., $\mathbf{a} \sim \pi_{\text{base}}(\cdot \mid s)$. The high-level decision state is the pair

$$x = (s, \mathbf{a}) \in \mathcal{X} := \mathcal{S} \times \mathcal{A}^H,$$

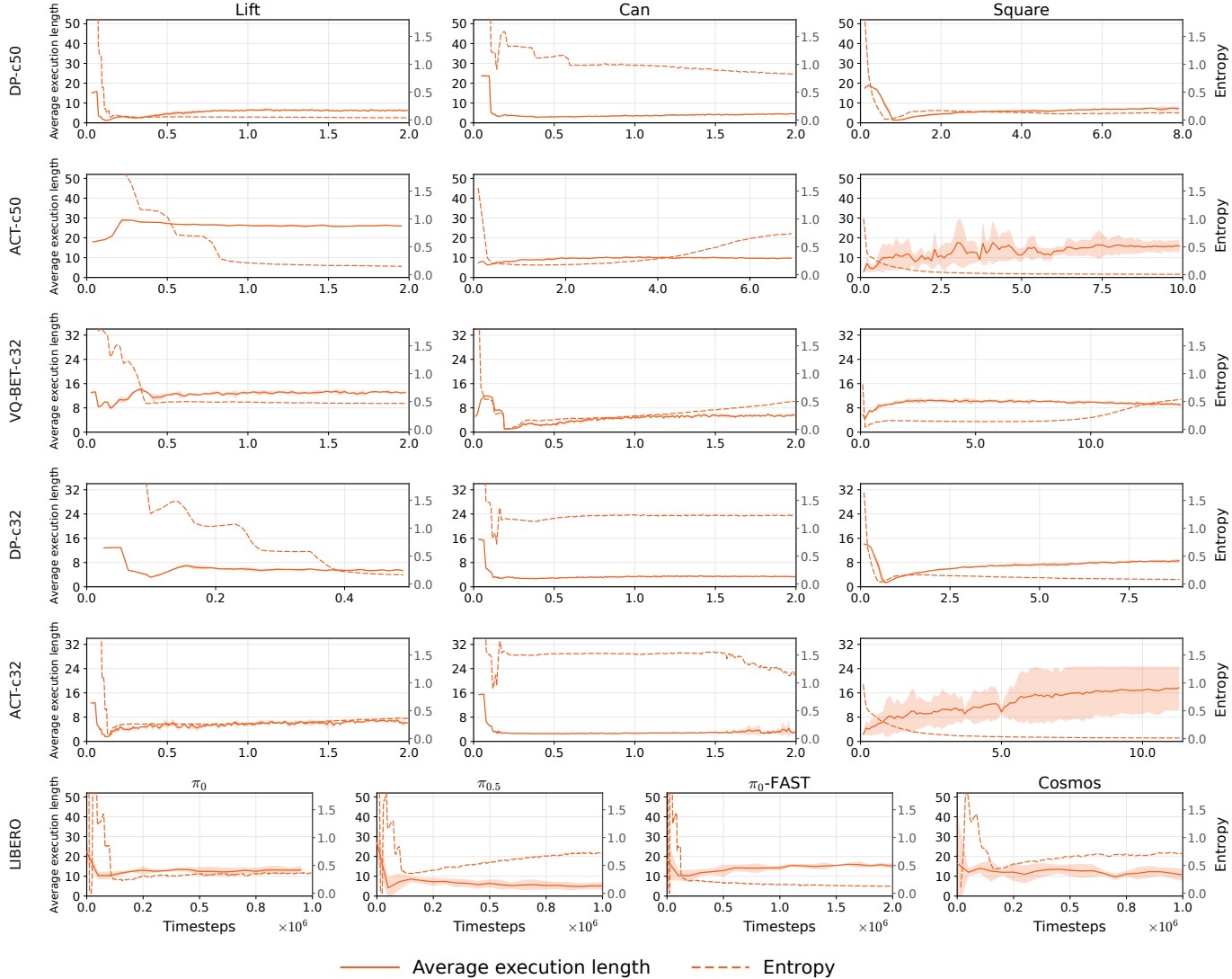

Fig. 9: **Evolution of execution lengths and policy entropy learned by ExRL-SAC.** We plot the average execution length selected by ExRL-SAC and the entropy of the execution-length policy during online adaptation. ExRL learns task- and policy-dependent execution-length patterns, while the entropy curves show how the policy gradually shifts from exploratory execution-length selection to more confident decisions.

and the high-level action is the execution length $k \in \mathcal{K} = \{1, \ldots, H\}$. Thus, ExRL induces an SMDP

$$\mathcal{M}_{\text{ex}} = (\mathcal{X}, \mathcal{K}, P_{\text{ex}}, r_{\text{ex}}, F_{\text{ex}}, \rho_{\text{ex}}, \gamma),$$

where $\mathcal{X}$ is the decision-state space, $\mathcal{K}$ is the execution-length action space, $P_{\text{ex}} : \mathcal{X} \times \mathcal{K} \to \Delta(\mathcal{X})$ is the transition kernel over execution-level decision states, $r_{\text{ex}} : \mathcal{X} \times \mathcal{K} \to \mathbb{R}$ is the reward function for one execution-length decision, $F_{\text{ex}} : \mathcal{X} \times \mathcal{K} \to \Delta(\mathbb{N})$ is the duration distribution over primitive timesteps, $\rho_{\text{ex}} \in \Delta(\mathcal{X})$ is the initial decision-state distribution, and $\gamma$ is the primitive-time discount factor inherited from the underlying MDP.

The initial decision-state distribution is induced by the en-

vironment initial state distribution and the frozen base policy:

$$s_0 \sim \rho_0, \qquad \mathbf{a}_0 \sim \pi_{\text{base}}(\cdot \mid s_0), \qquad x_0 = (s_0, \mathbf{a}_0) \sim \rho_{\text{ex}}.$$

Given a decision state $x = (s_t, \mathbf{a}_t)$ and an execution length $k_t$, the environment executes the prefix of the chunk for $k_t$ primitive timesteps:

$$a_t, a_{t+1}, \ldots, a_{t+k_t-1}.$$

This produces a trajectory segment $(s_t, a_t, r_t, \ldots, s_{t+k_t})$ under the primitive dynamics $P$. The reward of the high-level decision is the discounted reward accumulated along this segment,

$$r_{\text{ex}}(x, k_t) = \mathbb{E}\left[ \sum_{i=0}^{k_t-1} \gamma^i r(s_{t+i}, a_{t+i}) \,\middle|\, x, k_t \right].$$

| Task | $k_{\max}$ | Horizon | DDIM | $\gamma$ | Warmup transitions | Critic-only updates | LR | $N_Q$ | Target sub. | Critic MLP | Batch |
|---|---|---|---|---|---|---|---|---|---|---|---|
| Lift-32 | 32 | 300 | 5 | 0.997 | 512 | 5,000 | 3e−4 | 10 | 2 | 512×3 | 256 |
| Lift-50 | 50 | 300 | 5 | 0.997 | 512 | 5,000 | 3e−4 | 10 | 2 | 512×3 | 256 |
| Can-32 | 32 | 300 | 20 | 0.997 | 512 | 5,000 | 3e−4 | 10 | 2 | 512×3 | 256 |
| Can-50 | 50 | 300 | 20 | 0.997 | 512 | 5,000 | 3e−4 | 10 | 2 | 512×3 | 256 |
| Square-32 | 32 | 400 | 20 | 0.997 | 512 | 5,000 | 3e−4 | 10 | 2 | 512×3 | 256 |
| Square-50 | 50 | 400 | 20 | 0.997 | 512 | 5,000 | 3e−4 | 10 | 2 | 512×3 | 256 |
| Transport-32 | 32 | 800 | 20 | 0.997 | 512 | 5,000 | 3e−4 | 10 | 2 | 512×3 | 256 |

TABLE II: ExRL-Q hyperparameters for DP Robomimic experiments.

| Task | $k_{\max}$ | Horizon | $\gamma$ | Warmup transitions | Critic-only updates | LR | $N_Q$ | Target sub. | Critic MLP | Batch |
|---|---|---|---|---|---|---|---|---|---|---|
| Lift-32 | 32 | 300 | 0.997 | 512 | 5,000 | 3e−4 | 10 | 2 | 512×3 | 256 |
| Lift-50 | 50 | 300 | 0.997 | 512 | 5,000 | 3e−4 | 10 | 2 | 512×3 | 256 |
| Can-32 | 32 | 300 | 0.997 | 2,048 | 5,000 | 3e−4 | 10 | 2 | 512×3 | 256 |
| Can-50 | 50 | 300 | 0.997 | 512 | 5,000 | 3e−4 | 10 | 2 | 512×3 | 256 |
| Square-32 | 32 | 400 | 0.997 | 512 | 5,000 | 3e−4 | 10 | 2 | 512×3 | 256 |
| Square-50 | 50 | 400 | 0.997 | 512 | 5,000 | 3e−4 | 10 | 2 | 512×3 | 256 |

TABLE III: ExRL-Q hyperparameters for ACT Robomimic experiments.

| Task | $k_{\max}$ | Horizon | $\gamma$ | Warmup transitions | Critic-only updates | LR | $N_Q$ | Target sub. | Critic MLP | Batch |
|---|---|---|---|---|---|---|---|---|---|---|
| Lift-32 | 32 | 300 | 0.997 | 512 | 5,000 | 3e−4 | 10 | 2 | 512×3 | 256 |
| Can-32 | 32 | 300 | 0.997 | 512 | 5,000 | 3e−4 | 10 | 2 | 512×3 | 256 |
| Square-32 | 32 | 400 | 0.996 | 512 | 5,000 | 3e−4 | 10 | 2 | 512×3 | 256 |

TABLE IV: ExRL-Q hyperparameters for VQ-BeT Robomimic experiments.

| Model | $k_{\max}$ | Horizon | $\gamma$ | Warmup transitions | Critic-only updates | LR | $N_Q$ | Target sub. | Critic MLP | Batch |
|---|---|---|---|---|---|---|---|---|---|---|
| $\pi_0$ | 50 | 400 | 0.997 | 512 | 5,000 | 3e−4 | 10 | 2 | 512×3 | 256 |
| $\pi_{0.5}$ | 50 | 400 | 0.997 | 512 | 5,000 | 3e−4 | 10 | 2 | 512×3 | 256 |
| $\pi_0$-FAST | 50 | 250 | 0.997 | 512 | 3,000 | 3e−4 | 10 | 2 | 512×3 | 256 |
| Cosmos Policy | 50 | 250 | 0.997 | 512 | 5,000 | 3e−4 | 10 | 2 | 512×3 | 256 |

TABLE V: ExRL-Q hyperparameters for LIBERO experiments.

After the prefix execution ends, the base policy is queried again at the new state $s_{t+k_t}$ to produce the next chunk $\mathbf{a}' \sim \pi_{\text{base}}(\cdot \mid s_{t+k_t})$. Therefore, the next high-level decision state is

$$x' = (s_{t+k_t}, \mathbf{a}').$$

Let $K$ denote the realized duration of executing length $k_t$. Starting from $x = (s_t, \mathbf{a}_t)$, the primitive dynamics generate states $s_{t+1}, \ldots, s_{t+K}$ by executing the prefix of $\mathbf{a}_t$ until either $k_t$ actions have been executed or the episode terminates. The duration distribution is

$$F_{\text{ex}}(K \mid x, k_t) = \Pr(K \text{ primitive timesteps elapse}$$
$$\text{before the next decision} \mid x, k_t).$$

In the non-terminal case, $F_{\text{ex}}(K \mid x, k_t) = \mathbb{1}\{K = k_t\}$; if the episode terminates early, then $K < k_t$ is the actual number of primitive steps executed and no bootstrap term is used.

The transition kernel over decision states is

$$P_{\text{ex}}(x' \mid x, k_t) = \Pr(s_{t+K} = s', \ \mathbf{a}' \sim \pi_{\text{base}}(\cdot \mid s') \mid x, k_t).$$

That is, $P_{\text{ex}}$ is induced by first rolling out the primitive dynamics under the selected prefix, and then querying the frozen base policy at the resulting state to form the next decision state.

For an execution-length policy $\pi_{\text{ex}}(k \mid x)$, the SMDP Bellman equation is

$$Q^{\pi_{\text{ex}}}(x, k) = \mathbb{E}\left[R + \gamma^K V^{\pi_{\text{ex}}}(x') \mid x, k\right],$$

where

$$R = \sum_{i=0}^{K-1} \gamma^i r(s_{t+i}, a_{t+i}),$$

$$V^{\pi_{\text{ex}}}(x') = \mathbb{E}_{k' \sim \pi_{\text{ex}}(\cdot \mid x')}\left[Q^{\pi_{\text{ex}}}(x', k')\right],$$

| Task | $k_{\max}$ | Horizon | DDIM | $\gamma$ | Target entropy scale | LR (actor/critic/$\alpha$) | Warmup transitions | Actor&Critic MLP | Batch |
|---|---|---|---|---|---|---|---|---|---|
| Lift-32 | 32 | 300 | 5 | 0.999 | 0.98 | 3e−4/3e−4/1e−5 | 4000 | 512×3 | 1024 |
| Lift-50 | 50 | 300 | 5 | 0.999 | 0.98 | 3e−4/3e−4/1e−5 | 4000 | 512×3 | 1024 |
| Can-32 | 32 | 300 | 20 | 0.999 | 0.98 | 3e−4/3e−4/3e−4 | 4000 | 512×3 | 1024 |
| Can-50 | 50 | 300 | 20 | 0.999 | 0.95 | 3e−4/3e−4/3e−4 | 4000 | 512×3 | 1024 |
| Square-32 | 32 | 400 | 20 | 0.997 | 0.95 | 3e−4/3e−4/1e−5 | 4000 | 512×3 | 1024 |
| Square-50 | 50 | 400 | 20 | 0.997 | 0.95 | 3e−4/3e−4/1e−5 | 4000 | 512×3 | 1024 |
| Transport-32 | 32 | 800 | 20 | 0.999 | 0.95 | 3e−4/3e−4/1e−5 | 4000 | 512×3 | 1024 |

TABLE VI: ExRL-SAC hyperparameters for DP Robomimic experiments.

| Task | $k_{\max}$ | Horizon | $\gamma$ | Target entropy scale | LR (actor/critic/$\alpha$) | Warmup transitions | Actor&Critic MLP | Batch |
|---|---|---|---|---|---|---|---|---|
| Lift-32 | 32 | 300 | 0.997 | 0.85 | 3e−4/3e−4/3e−4 | 4000 | 512×3 | 1024 |
| Can-32 | 32 | 300 | 0.997 | 0.95 | 3e−4/3e−4/1e−5 | 4000 | 512×3 | 1024 |
| Square-32 | 32 | 400 | 0.997 | 0.85 | 3e−4/3e−4/1e−5 | 4000 | 512×3 | 1024 |

TABLE VII: ExRL-SAC hyperparameters for VQ-BeT Robomimic experiments.

| Task | $k_{\max}$ | Horizon | $\gamma$ | Target entropy scale | LR (actor/critic/$\alpha$) | Warmup transitions | Actor&Critic MLP | Batch |
|---|---|---|---|---|---|---|---|---|
| Lift-32 | 32 | 300 | 0.999 | 0.98 | 3e−4/3e−4/1e−5 | 4000 | 512×3 | 1024 |
| Lift-50 | 50 | 300 | 0.999 | 0.98 | 3e−4/3e−4/1e−5 | 4000 | 512×3 | 1024 |
| Can-32 | 32 | 300 | 0.999 | 0.98 | 3e−4/3e−4/3e−4 | 4000 | 512×3 | 1024 |
| Can-50 | 50 | 300 | 0.997 | 0.95 | 3e−4/3e−4/1e−5 | 4000 | 512×3 | 1024 |
| Square-32 | 32 | 400 | 0.999 | 0.80 | 3e−4/3e−4/3e−4 | 4000 | 512×3 | 1024 |
| Square-50 | 50 | 400 | 0.999 | 0.80 | 3e−4/3e−4/3e−4 | 4000 | 512×3 | 1024 |

TABLE VIII: ExRL-SAC hyperparameters for ACT Robomimic experiments.

| Model | $k_{\max}$ | Horizon | $\gamma$ | Target entropy scale | LR (actor/critic/$\alpha$) | Warmup transitions | Actor&Critic MLP | Batch |
|---|---|---|---|---|---|---|---|---|
| $\pi_0$ | 50 | 400 | 0.997 | 0.80 | 1e−4/3e−4/5e−5 | 1536 | 512×3 | 128 |
| $\pi_{0.5}$ | 50 | 400 | 0.997 | 0.80 | 1e−4/3e−4/3e−4 | 1536 | 512×3 | 128 |
| $\pi_0$-FAST | 50 | 250 | 0.997 | 0.80 | 1e−4/3e−4/3e−4 | 1536 | 512×3 | 128 |
| Cosmos Policy | 50 | 250 | 0.997 | 0.80 | 1e−4/3e−4/3e−4 | 1536 | 512×3 | 128 |

TABLE IX: ExRL-SAC hyperparameters for LIBERO experiments.

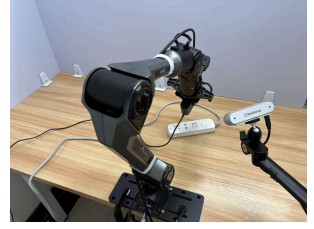
Power insertion task

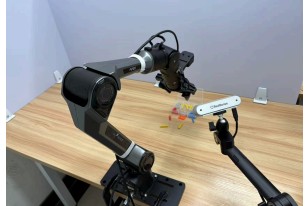
Parts organization task

Fig. 10: Real-world task setups.

and $K$ is the realized duration of the high-level execution-length decision. When the episode does not terminate during the prefix execution, $K = k$.

This recovers the duration-aware TD target used in ExRL. Writing $x = (s_t, a_{t:t+H})$, $x' = (s', a')$, and $K = k_t$, the target

becomes

$$y_t = \sum_{i=0}^{k_t-1} \gamma^i r(s_{t+i}, a_{t+i}) + \gamma^{k_t} V(s', a'),$$

where $V$ is instantiated by the particular RL algorithm, such as a greedy value for ExRL-Q or a soft value for ExRL-SAC. Thus, the only difference from a standard one-step Bellman backup is that the selected high-level action $k_t$ determines the duration of the transition. As a result, $k_t$ affects not only the accumulated reward and next decision state, but also the bootstrap discount through the factor $\gamma^{k_t}$.

### B. Evolution of Learned Execution Lengths

We visualize the average execution length selected by ExRL-SAC during online adaptation in Figure 9. We also plot the entropy of the execution-length policy, $H(\pi_{\text{ex}}(\ \cdot\ | s, a_{t:t+H})) = -\sum_{k \in \mathcal{K}} \pi_{\text{ex}}(\ k\ |\ s, a_{t:t+H}) \log \pi_{\text{ex}}(\ k\ | s, a_{t:t+H}))$, to measure how confident the policy is in its

| Setting | Power insertion | Parts organization |
|---|---|---|
| Language command | Insert the charger into the power strip | place the scattered pieces into the compartment box and close the lid |
| BC demonstrations | 128 (22,865 frames) | 21 (19,288 frames) |
| Base action chunk length | 50 | 50 |
| Warmup rollouts | 30 | 29 |
| Learning rate | 3e−4 | 3e−4 |
| Offline critic updates | 5,000 | 5,000 |
| Online updates per rollout | 400 | 400 |
| Maximum episode length | 450 steps | 1021 steps |
| Evaluation trials | 25 | 25 |

TABLE X: Real-world ExRL training settings. The base $\pi_{0.5}$ policies are frozen during ExRL adaptation; only the execution-length critic is trained.

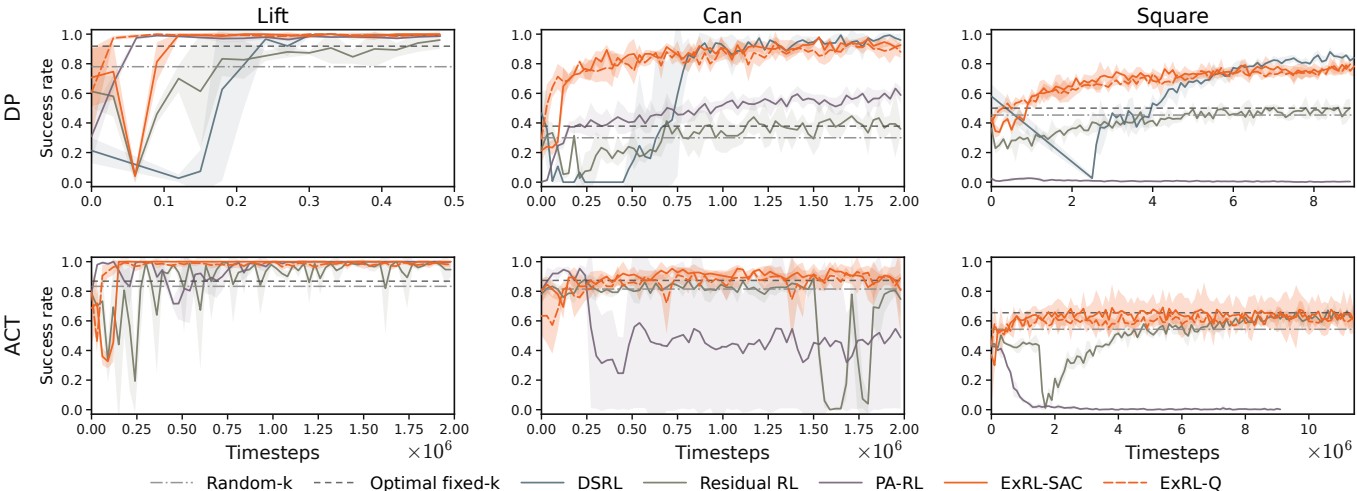

Fig. 11: Additional online adaptation results on `Robomimic` with base policy chunk size 32.

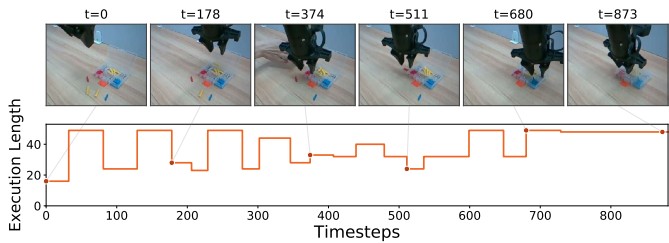

Fig. 12: **A representative rollout of ExRL on parts organization.** ExRL selects long execution lengths for confident pick-and-place segments, while adjusting the execution length across stages as the robot places parts into the storage box and closes it.

execution-length decisions. The learned execution length varies substantially across tasks and base policies, suggesting that it depends on both the task dynamics and the action chunks produced by the base policy.

A common trend is that the selected execution length often decreases early in training. This is expected because the replay buffer is still small and the critic is not yet well learned, so the backup is dominated by the immediate reward term. Since our reward uses a per-step penalty of $-1$, shorter executions can initially appear better, creating a shortcut toward small $k$. As the critic improves and the bootstrap term becomes more reliable, ExRL moves away from this early bias and learns task- and policy-specific execution-length patterns.

The entropy curves further help explain when ExRL behaves adaptively and when it degenerates to a nearly fixed execution length. In successful cases, the policy often maintains nonzero entropy or gradually settles after learning useful state- and chunk-dependent decisions. In contrast, as shown in Figure 2 and 11, on ACT `Square`, ExRL-SAC does not outperform the optimal fixed-$k$ baseline. We see that in this case, the entropy quickly drops close to zero. This indicates that the learned execution-length policy collapses to an almost deterministic choice, making it behave similarly to a fixed-$k$ policy. This suggests that ExRL's advantage over fixed execution lengths depends on learning genuinely adaptive replanning behavior.

### C. Experimental Details

*1) ExRL-Q Hyperparameters:* Tables II–V summarize the standard ExRL-Q hyperparameters used in our plotted ex-

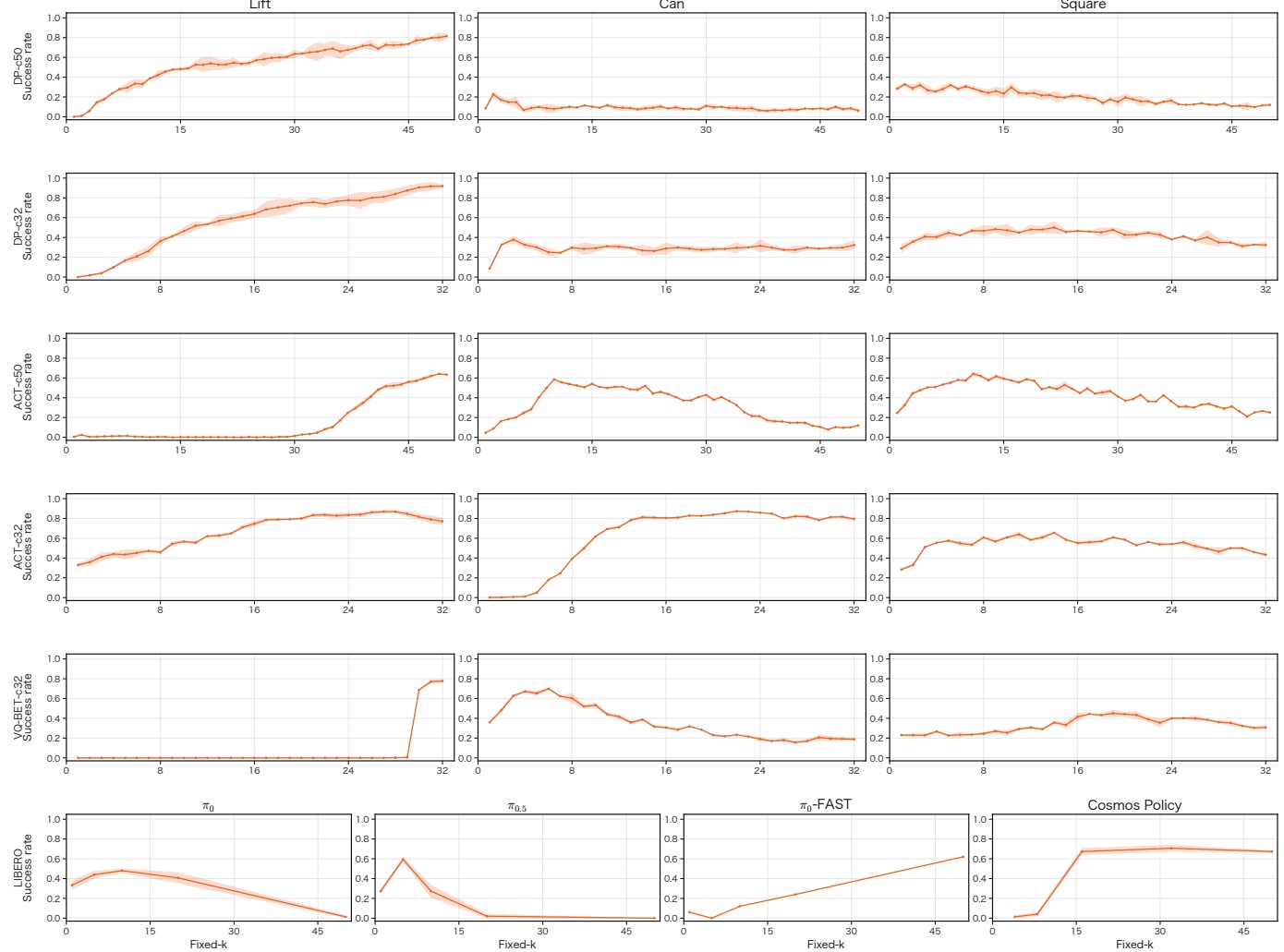

Fig. 13: **Fixed-$k$ sweep across base policies and tasks.** We evaluate each base policy under different fixed execution lengths. Performance varies substantially with $k$, and the optimal fixed execution length differs across tasks and policies, highlighting the importance of adaptive execution-length control.

periments. For DP base policies, DDIM denotes the number of denoising steps used when querying the frozen policy at inference time; this column is omitted for ACT and VQ-BeT because their policy inference is not DDIM-based. Warmup transitions are collected with uniformly random execution lengths before any critic update. Critic-only updates are gradient steps on this warmup replay before online critic-guided rollout starts. LR is the learning rate for the critic. $N_Q$ is the critic ensemble size, Target sub. is the number of target critics sampled for each Bellman backup, Critic MLP reports the critic hidden-layer structure and Batch reports the minibatch size.

*2) ExRL-SAC Hyperparameters:* Tables VI-IX summarize the standard ExRL-SAC hyperparameters used in our plotted experiments. DDIM denote the number of inference steps used by the frozen base policy. Target entropy scale multiplies the default discrete-action SAC target entropy used for automatic temperature tuning. LR lists the actor, critic, and temperature

learning rates, respectively; Actor&Critic MLP reports the hidden-layer width/depth and the SAC batch size.

*3) Details of Real-World Experiment:* All real-world experiments use a agilex Piper arm controlled at 20Hz. The frozen base policy is a $\pi_{0.5}$ policy fine-tuned with behavior cloning on task-specific human demonstrations. At each query, the base policy receives two RGB camera views (a third-person camera and a wrist camera) together with the robot proprioceptive state, and predicts an action chunk of length $H = 50$. ExRL does not change the predicted actions or update the base policy. Instead, it observes the current robot observation and the predicted action chunk, and selects an execution length $k \in \{1, \ldots, 50\}$. The robot executes the first $k$ primitive actions, after which the base policy is queried again. We use primitive discount $\gamma = 0.997$.

We evaluate two real-world tasks, each using a separate $\pi_{0.5}$ base policy and a separate task-specific behavior-cloning dataset. In *power insertion*, the robot must insert a charger plug

into a power strip. We use the language instruction "Insert the charger into the power strip" and train the base policy from 128 demonstrations (22,865 frames). We treat this as a single-stage success task: each rollout succeeds when the plug is inserted and otherwise terminates at the horizon. In *parts organization*, the robot must place several different small parts into a compartment storage box and then close the lid. We use the language instruction "place the scattered pieces into the compartment box and close the lid" and train the base policy from 21 demonstrations (19,288 frames). This task has five progress stages: four successful part placements and one final lid-closing stage. During real-world interaction, the operator only provides sparse event annotations: for power insertion, the operator labels the trial as success or failure; for parts organization, the operator presses a key whenever a stage is completed. The operator does not provide action corrections or teleoperation during RL.

**Reward and online training protocol.** For power insertion, we use a sparse time-to-success reward. Each primitive timestep before success receives reward $-1$, the terminal success timestep receives reward $0$, and failed rollouts terminate at the horizon. For parts organization, we use a sparse stage-completion reward: completing a stage gives reward $+1$ at the corresponding primitive timestep, while all other timesteps have zero reward. We initialize ExRL with random execution-length exploration before online adaptation. For power insertion, we collect 30 random-$k$ warmup rollouts, producing $419$ transitions (10,513 primitive robot steps). For parts organization, we use roughly the same warmup scale, collecting 29 random-$k$ rollouts. We then train the ExRL-Q critic for 5,000 critic-only updates on the warmup replay. During online adaptation, each new real-world rollout is inserted into replay after receiving the sparse human annotation, and the critic is updated for 400 gradient steps before the next rollout. The critic uses an ensemble of 10 Q-functions with mean reduction, hidden dimensions $(1024, 1024, 1024)$, batch size 64, and target critic subsampling size 2. We continue online adaptation for at most a few hundred real-world rollouts; the reported power insertion run used 142 total rollouts, and the parts organization run used 250 total rollouts. After training, we freeze the learned execution-length critic and evaluate for 25 trials per task.