# OpenReview forum: "ExRL: Sample-Efficient Online Adaptation of Action Chunking Policies via Execution Length Control"
_roboticsfoundation.org/RSS/2026/Workshop/RL4VLA — RL4VLA_

### Official Review · Reviewer_uugd · 2026-06-27

**Rating:** 4
**Confidence:** 4

**Review:**

## Summary

The paper proposes ExRL, a lightweight online RL method for adapting frozen action-chunking robot policies by learning how many actions from each predicted chunk should be executed before replanning. Instead of modifying high-dimensional action chunks or fine-tuning the base policy, ExRL learns a discrete execution-length controller conditioned on the current observation and the predicted action chunk. The method is instantiated as ExRL-Q and ExRL-SAC, both using a duration-aware SMDP-style Bellman backup. The paper evaluates ExRL on Robomimic, LIBERO with several generalist robot policies, and two real-world manipulation tasks.

## Strengths

1. The motivation is strong. Fixed execution length is a practical bottleneck for action-chunking policies, especially when deploying VLA models in contact-rich or long-horizon manipulation tasks.

2. The proposed method is policy-agnostic and relatively simple to implement. Since the base policy is frozen, ExRL provides a lightweight adaptation mechanism without directly fine-tuning large VLA or flow-based policies.

3. The paper includes real-world experiments, and the reported results are promising.

## Weaknesses

1. Some very closely related baselines are missing. In particular, the paper does not compare against recent adaptive action-chunking methods such as AAC, which also adjusts chunk sizes automatically and is training-free. Since ExRL’s central contribution is learning when to replan within an action chunk, comparisons with adaptive chunk execution methods are important for establishing the empirical advantage of online RL over simpler inference-time adaptation strategies.

2. From an RL perspective, the method is closely related to prior work on temporal abstraction and semi-MDP decision making, such as TempoRL and FiGAR, where an agent learns how long to commit to an action or behavior using Q functions. This does not diminish the novelty of ExRL, since they were mostly developed in RL settings rather than VLA models. However, the paper should more clearly discuss the conceptual relationship and differences between ExRL and such methods.

3. When the base policy is changed, does the learned ExRL controller still remain applicable? If it needs to be retrained, does this introduce additional data and computational requirements in practical deployment compared with training-free methods such as AAC and FFDC?

## References

[1] A. Biedenkapp, R. Rajan, F. Hutter, and M. Lindauer, “TempoRL: Learning When to Act,” in Proceedings of the 38th International Conference on Machine Learning (ICML), PMLR, vol. 139, pp. 914–924, 2021.

[2] S. Sharma, A. S. Lakshminarayanan, and B. Ravindran, “Learning to Repeat: Fine Grained Action Repetition for Deep Reinforcement Learning,” in International Conference on Learning Representations (ICLR), 2017.

[3] Y. Liang, X. Wang, K. Wang, S. Wang, X. Peng, H. Chen, D. K. H. Chua, and P. Vadakkepat, “Adaptive Action Chunking at Inference-time for Vision-Language-Action Models,” in Proceedings of the IEEE/CVF Conference on Computer Vision and Pattern Recognition (CVPR), 2026.

[4] R. Wang, Y. Zhang, J. Lin, K. Luo, J. Wang, Z. Wang, and X. Qi, “When to Trust Imagination: Adaptive Action Execution for World Action Models,” arXiv preprint arXiv:2605.06222, 2026.

---

### Official Review · Reviewer_Y4b7 · 2026-06-29

**Rating:** 6
**Confidence:** 3

**Review:**

# Review: ExRL

This paper proposes ExRL, a lightweight online RL adapter for action-chunking robot policies. Instead of fine-tuning the high-dimensional action generator, ExRL freezes the base policy and learns a discrete execution-length policy deciding how many actions from each predicted chunk to execute before replanning.

**Pros**

- Simple and well-motivated adaptation interface: optimizing execution length is much lower-dimensional than optimizing action chunks.
- Strong empirical breadth: Robomimic, LIBERO, multiple base policy families, and two real-world tasks.
- Good comparison set, including fixed/random execution lengths, residual RL, PA-RL, and DSRL.
- The duration-aware backup and chunk-conditioned execution policy are clean technical details, and the ablations support them.
- Real-world results are compelling, especially for precision insertion and long-horizon parts organization.

**Cons**

- The method is bounded by the frozen base policy's action support; this is acknowledged, but it limits applicability when the base chunks are systematically wrong.
- Real-world evaluation has limited statistical treatment; confidence intervals or multiple seeds/training runs would make the claims stronger.
- Some novelty claims should be softened given closely related concurrent adaptive chunk-execution work.
- The experiments focus on single-task adaptation; the implications for language-conditioned multi-task RL remain unclear.
- More detail on reward design, reset conditions, safety constraints, and online interaction cost would help reproducibility.

---

### Decision · Program_Chairs · 2026-07-03

**Decision:**

Accept

**Comment:**

This paper proposes ExRL, a lightweight test-time method that learns how many actions from a predicted action chunk should be executed before predicting the next chunk while keeping the policy frozen. The reviewers appreciated the motivation and the experimental results, with the main concerns being the missing related work and a lack of experimental details. We agree that these are valid concerns, but we do not consider them sufficient to reject the paper. We encourage the authors to address these issues in the camera-ready version, particularly by discussing relevant training-free methods such as AAC and FFDC, and by providing additional experimental details to improve reproducibility.